computational biology/biophysics

arrhythmia, electrophysiology, computational biology, pharmacology

**Author for correspondence:**
Rebecca-Ann B. Burton
e-mail: rebecca.burton@pharm.ox.ac.uk

†Joint senior authors.

# Cardiac TdP risk stratification modelling of anti-infective compounds including chloroquine and hydroxychloroquine

Dominic G. Whittaker[1], Rebecca A. Capel[2], Maurice Hendrix[1,3], Xin Hui S. Chan[4,5], Neil Herring[6], Nicholas J. White[4,5], Gary R. Mirams[1,†] and Rebecca-Ann B. Burton[2,†]

[1]Centre for Mathematical Medicine and Biology, School of Mathematical Sciences, University of Nottingham, Nottingham, UK
[2]Department of Pharmacology, University of Oxford, Oxford, UK
[3]Digital Research Service, University of Nottingham, Nottingham, UK
[4]Mahidol-Oxford Tropical Medicine Research Unit, Mahidol University, Bangkok, Thailand
[5]Centre for Tropical Medicine and Global Health, Nuffield Department of Medicine, University of Oxford, Oxford, UK
[6]Department of Physiology, Anatomy and Genetics, University of Oxford, Oxford, UK

DGW, 0000-0002-2757-5491; MH, 0000-0002-6621-7996;
XHSC, 0000-0002-9941-6975; GRM, 0000-0002-4569-4312;
R-ABB, 0000-0002-0904-3862

Hydroxychloroquine (HCQ), the hydroxyl derivative of chloroquine (CQ), is widely used in the treatment of rheumatological conditions (systemic lupus erythematosus, rheumatoid arthritis) and is being studied for the treatment and prevention of COVID-19. Here, we investigate through mathematical modelling the safety profile of HCQ, CQ and other QT-prolonging anti-infective agents to determine their risk categories for *Torsade de Pointes* (TdP) arrhythmia. We performed safety modelling with uncertainty quantification using a risk classifier based on the qNet *torsade metric score*, a measure of the net charge carried by major currents during the action potential under inhibition of multiple ion channels by a compound. Modelling results for HCQ at a maximum free therapeutic plasma concentration (free $C_{max}$) of approximately 1.2 μM (malaria dosing) indicated it is most likely to be in the high-intermediate-risk category for TdP, whereas CQ at a free $C_{max}$ of approximately 0.7 μM was predicted to most likely lie in the intermediate-risk category. Combining HCQ with the

antibacterial moxifloxacin or the anti-malarial halofantrine (HAL) increased the degree of human ventricular action potential duration prolongation at some or all concentrations investigated, and was predicted to increase risk compared to HCQ alone. The combination of HCQ/HAL was predicted to be the riskiest for the free $C_{max}$ values investigated, whereas azithromycin administered individually was predicted to pose the lowest risk. Our simulation approach highlights that the torsadogenic potentials of HCQ, CQ and other QT-prolonging anti-infectives used in COVID-19 prevention and treatment increase with concentration and in combination with other QT-prolonging drugs.

# 1. Introduction

The cinchona alkaloid quinine (QUIN), along with synthetically produced chloroquine (CQ), are quinoline compounds which have been used in the treatment of malaria for decades. The more soluble hydroxy derivative of CQ is hydroxychloroquine (HCQ). CQ and HCQ are diprotic bases which accumulate in acid vesicles including lysosomes over time. Many of their multiple biological activities including their antiviral action are associated with increased vesicle pH [1]. Both drugs have been shown to block various cardiac ion channels [2].

HCQ was initially developed as an anti-malarial drug, sold as the sulfate salt under the trade name Plaquenil [3]. HCQ is used for the treatment of a wide variety of conditions with the majority of use being for systemic lupus erythematosus (SLE) and rheumatoid arthritis (RA). For these indications, it is often prescribed for use over months to years, and has had a good safety record including in pregnancy [1].

Early clinical studies reported few toxic side effects across HCQ anti-malarial treatment regimens [4]. Work in the late 1950s [5] explored the possibility of using 4-aminoquinolines in the treatment of cardiac arrhythmias, although the specific action of reducing heart rate was not explored. Sumpter *et al.* [6] showed evidence for risk of cardiomyopathy during long-term exposure to high doses of HCQ for the treatment of patients with SLE and RA. The present treatment regime for SLE (generally 200–400 mg d$^{-1}$) [7] includes doses lower than the one originally used to treat arthritis or malaria [8]. Irreversible retinal toxicity is rare at current recommended doses [9,10]. Reported side effects have been seen in greater than 1000 mg d$^{-1}$ dosage ranges [11–13]. The risk of retinopathy is increased with large cumulative doses of HCQ (greater than 1000 g) [14]. The arrhythmogenic cardiotoxicity of the quinoline and structurally related anti-malarial drugs are well documented [15]; in particular, effects on hypotension and electrocardiographic QT interval prolongation have been reported. Capel *et al.* [16] in 2015 showed that HCQ also inhibits the pacemaking current $I_f$ and offers the potential of being used as a bradycardic agent. They also noted additional effects on the *L-type calcium channels* and *delayed rectifier potassium channels* in isolated guinea pig sinoatrial node cells, indicating multi-ion channel block in cardiac cells. With the increased re-purposing of CQ and HCQ [14], including for the treatment (SOLIDARITY trial, ISRCTN83971151 and UK RECOVERY trial, ISRCTN50189673) and prevention of COVID-19, there is a need to critically assess the cardiovascular safety profiles of these anti-malarials.

Since the advent of mathematical modelling of cardiac cell activity in the 1960s [17], major new insights have emerged in the field, along with approaches to calibrating such models from experimental data [18], and recognition of the need to quantify uncertainty in predictions [19]. Mathematical modelling has since been shown to be useful in elucidating the requirements for reliable risk assessment predictions, such as the need to account for the actions of compounds on multiple ion channels [20], and has helped to guide experimental design considerations for ion channel screening experiments [21,22].

Prolongation of the QT interval on the surface electrocardiogram (ECG) is a surrogate measurement of prolonged ventricular action potential duration (APD). Dispersion of repolarization (DR) is a result of heterogeneous lengthening of APD throughout the ventricular myocardium, often across the ventricular walls. This DR and the tendency of prolonged APD associated with early afterdepolarizations (EADs) provide the substrate of polymorphic ventricular tachyarrhythmia (VT) associated with long QT syndrome (LQTS) [23], *Torsade de Pointes* (TdP) VT [24]. The vast majority of acquired LQTS cases (which are more prevalent than congenital LQTS cases) are the results of electrolyte abnormalities [25] or adverse drug effects [26], the latter particularly due to interaction with the human Ether-à-go-go-Related Gene (hERG), which encodes the pore-forming subunits (Kv11.1) of the channel carrying the rapidly activating delayed rectifier current, $I_{Kr}$.

In drug development, hERG IC$_{50}$ value estimates are used in the pre-clinical assessment of TdP risk. Redfern *et al.* [27] proposed [hERG IC$_{50}$/EFTPC$_{max}$] as an improvement over the simplified [hERG IC$_{50}$].

Mirams *et al*. [20] showed a simulated evaluation of multi-channel effects at the whole-cell level could be used to improve this early prediction of TdP risk. The comprehensive *in vitro* proarrhythmia assay (CiPA) initiative was later established as a novel cardiac safety screening paradigm that takes into account multi-channel drug effects intended to replace the former regulatory strategy which relied on hERG block and QT prolongation, sensitive predictors of TdP which lack specificity [28].

In patients with compromised organ function, such as in COVID-19, understanding drug safety and drug interactions is critical. In this study, we use a computational approach, based on a classifier developed by the CiPA initiative [29], to predict the clinical torsadogenic risk categories associated with CQ, HCQ and other commonly used anti-infective agents known to prolong the QT interval. We also use an independent block model to assess the safety profile of combination therapies and show that torsadogenic potentials of HCQ, CQ and other QT-prolonging anti-infectives used in COVID-19 prevention and treatment increase with concentration and in combination with other QT-prolonging drugs. Although the interest in CQ and HCQ for COVID-19 prophylaxis/treatment has waned, it is hoped that our approach may be considered for the screening of potential future therapies/combination therapies.

## 2. Methods

In 2015, we showed that HCQ inhibits $I_f$, L-type calcium channels, and slow and rapid delayed rectifier potassium channels in isolated guinea pig sinoatrial node cells [16]. Five minutes of exposure to 3 μM HCQ conveyed a statistically significant reduction in $I_{CaL}$ ($12 \pm 4\%$ reduction in max. conductance, $n = 6$) and $I_{Kr}$ ($35 \pm 4\%$ reduction across step lengths rendering maximal current activation, $n = 5$) at $p < 0.05$, analysed using repeated-measures ANOVA. We used these data to approximate $IC_{50}$s for the actions of HCQ on $I_{Kr}$, $I_{CaL}$ and $I_{Ks}$ by fitting the parameters of a Hill curve (assuming a Hill coefficient of 1 [20]). HCQ blocked channels in the following order of potency: $I_{Kr} > I_{Ks} > I_{CaL}$. $IC_{50}$ values for other anti-infective compounds of interest including azithromycin (AZ), chloroquine (CQ), halofantrine (HAL), lopinavir/ritonavir (LOP/RIT), moxifloxacin (MOX) and QUIN simulated in this study were taken from the literature, and are shown in electronic supplementary material, table S1.

Based on the availability of ion channel block data for each compound, changes in up to seven ion currents, namely $I_{Kr}$, $I_{Ks}$, $I_{K1}$, $I_{CaL}$, $I_{Na}$, $I_{NaL}$, $I_{to}$, were inputted into the CiPA version of the O'Hara-Rudy (ORd) human ventricle mathematical action potential model [29]. For example, based on the values presented in electronic supplementary material, table S1, AZ was assumed to block $I_{Kr}$, $I_{NaL}$, $I_{Ks}$ and $I_{to}$ with $IC_{50}$s of 70.8, 189.1, 470.0 and 88.8 μM, respectively. Drug block was modelled using conductance block where a proportion $b_i$ of channel type $i$ are blocked, and the maximal density of the current is then scaled by $(1 - b_i)$ [20]. For simulation of combinations of drugs, the Bliss model of independent block was assumed [30], in which the total proportion of channels blocked arising from the combination of the block by drugs 1 and 2 was given by

$$b_i = b_{1,i} + b_{2,i} - b_{1,i}b_{2,i}. \tag{2.1}$$

That is, block occurs when one compound, the other, or both are bound to an individual channel, and any of these scenarios leads to complete block of an individual channel. Combinations of drugs were applied at plasma concentrations based on the ratio of free $C_{max}$ for drugs from clinical studies referenced in electronic supplementary material, table S1 (where a range is shown, the highest available $C_{max}$ was used). A torsade metric score was calculated for each compound, computed as the average qNet at $1–4\times C_{max}$ [28]. Briefly, qNet is a measure of the net charge crossing the membrane during a simulated action potential repolarization, calculated as the integral or area under the curve of a net current, $I_{Net}$, defined as

$$I_{Net} = I_{CaL} + I_{NaL} + I_{Kr} + I_{Ks} + I_{K1} + I_{to}. \tag{2.2}$$

A previous study found that estimates of $pIC_{50}$ from repeated ion channel screens followed a logistic form, i.e. $pIC_{50} \sim \text{logistic}(\mu, \sigma)$ [21], where $\mu$ is the mean and $\sigma$ is a spread parameter. Obtaining a value for $\sigma$ in each of the affected ion channels allows uncertainty in $IC_{50}$ estimates to be propagated through to APD and qNet predictions. We used previously reported values of $\sigma$ [21] for $I_{Kr}$, $I_{Ks}$, $I_{Na}$, $I_{CaL}$ and $I_{to}$, or a conservative value of 0.15 for ion channels for which this information was not available ($I_{NaL}$ and $I_{K1}$). A $pIC_{50}$ estimate for each channel of interest was subsequently sampled from the corresponding logistic distribution—a process which was repeated 1000 times for each compound, resulting in 1000 distinct concentration–APD and concentration–qNet response curves. Ignoring the upper and lower 5% of

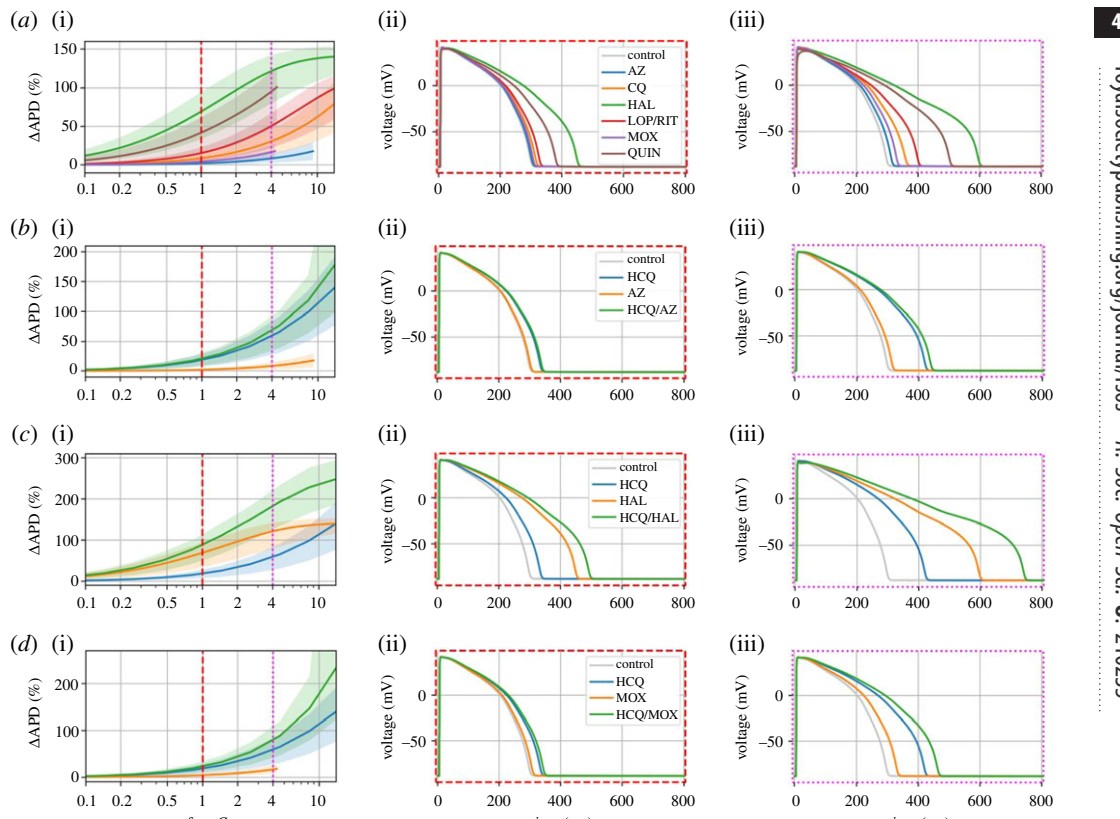

**Figure 1.** (a)(i) APD–concentration curves with 90% credible intervals from 1000 samples for control and azithromycin (AZ), chloroquine (CQ), halofantrine (HAL), lopinavir/ritonavir (LOP/RIT), moxifloxacin (MOX) and quinine (QUIN), with selected APs at concentrations of 1× free $C_{max}$ (ii) and 4× free $C_{max}$ (iii). Combinations of hydroxychloroquine (HCQ) with AZ, HAL and MOX are shown in (b), (c) and (d), respectively.

outputs allowed us to obtain an estimate of the 90% credible interval for simulated response curves. All AP simulations and qNet calculations were performed using ApPredict, a bolt-on extension to Chaste (which also has a web-portal front end [31]). All simulation data and codes required to run the simulations are freely available in the Github repository: https://github.com/CardiacModelling/risk-stratification-anti-malarials.

## 3. Results

In this study, we used increases in the cellular APD as a surrogate for QT interval prolongation [20]. The effects of each of the drugs on the APD were tested at log-spaced concentrations ranging from 0.001 to 100 μM and plotted in terms of the free $C_{max}$, in order to measure the % change in $APD_{90}$ with concentration. The $APD_{90}$ as a function of concentration is shown for AZ, CQ, HAL, LOP/RIT, MOX and QUIN in figure 1a, with APs at concentrations equivalent to 1× and 4× free $C_{max}$ highlighted. The compounds investigated generally had a free $C_{max}$ which is much less than the hERG $IC_{50}$ and so produced only a small degree of APD prolongation at 1× free $C_{max}$, including lopinavir/ritonavir. HAL on the other hand produced substantial APD prolongation at 1× free $C_{max}$ due to comparable values of hERG $IC_{50}$ and free $C_{max}$. QUIN produced smaller but still substantial APD prolongation at 1–4× free $C_{max}$. At a concentration 4× free $C_{max}$, LOP/RIT and CQ also produced fairly substantial APD prolongations, whereas this remained comparatively small for AZ and MOX.

Figure 1b–d shows APD-concentration curves for HCQ and various drugs administered alone, and in combination with HCQ. While the antibiotics AZ and MOX both had very minor APD prolonging effects when administered alone, they were both predicted to increase the overall degree of APD prolongation slightly when given with HCQ compared to HCQ alone. Both HAL and HCQ had a considerable effect when administered individually, so produced a substantial APD prolongation when combined (especially at 4× free $C_{max}$).

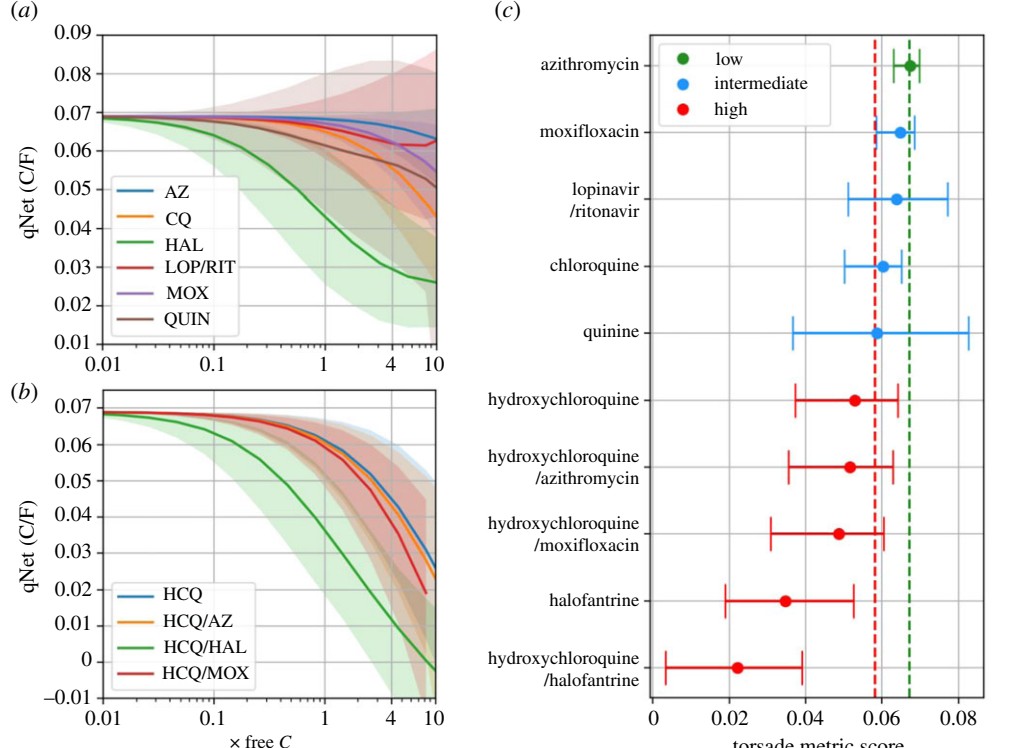

**Figure 2.** qNet–concentration curves with 90% credible intervals from 1000 samples for (*a*) azithromycin (AZ), chloroquine (CQ), halofantrine (HAL), lopinavir/ritonavir (LOP/RIT), moxifloxacin (MOX) and quinine (QUIN), and (*b*) hydroxychloroquine (HCQ), HCQ/AZ, HCQ/HAL and HCQ/MOX. (*c*) Torsade metric scores for individual compounds and combinations, separated into low, intermediate and high risk based on thresholds in [28], shown as red and green dotted lines.

Figure 2 shows the concentration-qNet curves for individual compounds and drug combinations, as well as associated torsade metric scores. Based on APD prolongation at a concentration of 1–4× free $C_{max}$ in figure 1, one would expect lopinavir/ritonavir to be slightly riskier than CQ. The qNet for CQ at this concentration, however, was lower (riskier) than for lopinavir/ritonavir (figure 2*a*), highlighting the different information provided by qNet; namely, that the balance of currents leading to APD prolongation is altered in a riskier way for CQ.

AZ was predicted to have the safest (highest) median qNet, yet the uncertainties around lopinavir/ritonavir and QUIN were greater, so they could be the safer compounds/combination. Lopinavir and ritonavir have almost identical hERG $IC_{50}$s, which also coincide somewhat with the $I_{CaL}$ and $I_{NaL}$ $IC_{50}$s for ritonavir, all of which would be expected to influence the degree of APD prolongation in the model. Therefore, as the concentration approaches the hERG (and other) $IC_{50}$ values there were several inputs close to the region of maximum uncertainty in the dose-response curve, generating large uncertainty in the output.

Combinations of drugs with HCQ decreased qNet in the order HAL > MOX > AZ (figure 2*b*). The torsade metric scores shown in figure 2*c* reveal that HCQ/HAL was predicted to be the riskiest combination of drugs, whereas HAL was predicted to be a highly risky individual drug. At the other end of the scale, AZ was the only compound placed in the low-risk category. However, combining AZ with HCQ resulted in a most likely high-risk outcome. CQ was placed in the intermediate-high-risk category, compared to mostly high risk for HCQ. As a free $C_{max}$ for HCQ treatment for which plasma binding was taken into account was not available for SLE and/or COVID doses, we used a value associated with a malaria dose [32].

## 4. Discussion

In this study, we used a previously developed classifier [29] to predict the clinical torsadogenic risk category associated with commonly used anti-malarials and anti-infective agents which prolong the QT interval, as well as an independent block model to assess the safety profile of combination

therapies that block multiple ion channels. These simulations were based on the earliest pre-clinical data available (measurement of $IC_{50}$s, typically from high-throughput assays), and integrate multi-channel effects to better predict risk classifications of drugs. Drugs such as CQ and HCQ have garnered a lot of attention over many years as they appear to work via multiple mechanisms and hence have shown promise in several disease conditions. Most recently, significant interest in the compounds was shown regarding the treatment and prevention of COVID-19. This interest has waned after cessation of the HCQ arm of the RECOVERY trial [33] and lack of impact, combined with the potential for adverse events [34,35], although support for the use of HCQ, AZ and their combined use continues to appear in recent publications in both the peer-reviewed [36,37] and popular literature [38] and discussions continue as to the merits of adjuvant therapy with zinc supplementation [39,40].

Drug safety requires the ability to predict and assess risk with an acceptable level of certainty including on major target organs [41]. We have taken advantage of recent developments in computational modelling [21,29] and knowledge derived from single-cell electrophysiological measurements of different quinolines (including CQ, HCQ, quinidine, QUIN and HAL) on up to seven major ion currents including $I_{Kr}$ and $I_{CaL}$ to stratify the risk of some of these compounds both individually and in combination with other compounds. Furthermore, as such models require increased scrutiny for use in safety-critical applications [19], we have performed uncertainty quantification. Our modelling showed that combining HCQ with AZ prolonged APD to a greater extent than the use of either individually (albeit this was notable only at 4× free $C_{max}$), and placed this combination in the high-risk category associated with measurable/unacceptable incidence of TdP. Furthermore, combining HCQ with MOX or HAL was also predicted to increase TdP risk.

Data in animal models have previously shown HCQ to have multi-channel actions including on calcium and potassium currents [16]. It is known that concurrent block of calcium currents with $I_{Kr}$ inhibition can protect against TdP and our modelling data suggest this is indeed the case for many compounds, including HCQ at low concentrations. Modelling results for a free $C_{max}$ of approximately 1.2 µM indicated that the balance of potassium and calcium current inhibitions observed would not be expected to lengthen the human ventricular action potential significantly at a dose of 1× free $C_{max}$ (figure 1), but did so at 4× free $C_{max}$, which resulted in an overall high-intermediate-risk category for HCQ. However, we also showed that accounting for variability in our experimental data led to a range of possibilities. With the exception of HAL and HCQ/HAL, all compounds and combinations investigated produced credible intervals for the torsade metric score that spanned at least two risk categories. Gathering more high-quality experimental data regarding the multi-channel inhibition effects of HCQ and other anti-malarials would allow us to predict with more confidence the most plausible APD prolongation range and risk category. We could also have accounted for different Hill coefficients, although it has been noted previously that the associated level of variability in their measurement is so high that it is unclear whether including the Hill coefficient from ion channel screening adds useful information [21]. It should be noted, in addition, that the $IC_{50}$ used as a model input has limitations as a measure of drug block, in that its value may depend on the electrophysiology protocol used [42,43], and, relatedly, it is unable to account for dynamic, state-dependent effects of a compound. Recent studies have integrated dynamic effects into computational models through the use of drug-binding kinetic schemes with rates inferred from specialized electrophysiology protocols [29] and atomistic scale measurements [44].

Our modelling predicted that CQ at a free $C_{max}$ of approximately 0.7 µM was generally safer than HCQ at a free $C_{max}$ of approximately 1.2 µM (used in the treatment of malaria at 400 mg [32]; figure 2). However, it should be noted that the free $C_{max}$ for HCQ is approximately 2 times greater than that for CQ; if the same free $C_{max}$ is used for both compounds then the risk scores are comparable (electronic supplementary material, figure S1). High doses of CQ are thus still expected to pose a high risk. Furthermore, the free $C_{max}$ we used for HCQ was for a malaria dose, which is a higher dose than for SLE. We expect that HCQ at the lower doses used for SLE thus remains generally safe (towards the intermediate-risk category), whereas higher doses used for COVID-19 will place HCQ unequivocally in the high-risk category (electronic supplementary material, figure S1). This is in keeping with clinical experience where CQ and HCQ are known to cause sudden death in overdose but have an otherwise good safety record in the treatment of malaria and SLE, respectively [15].

The risk categories presented in this study are based on a combination of adverse events, case reports and clinical judgement by an expert panel, more information about which can be found on the CiPA website [45]. The categories should not be interpreted as the risk of developing TdP when taking a particular compound (which would suggest that someone taking HCQ is at high risk of developing TdP), but rather the likelihood with which TdP that arises in a patient (which may remain extremely

rare) can be attributed to a particular compound. To put some of the risk categories in context, they are compared to quinidine (which has known TdP risk) in electronic supplementary material, figure S2. The torsade metric score (correctly) identified quinidine as high risk, suggesting that it is highly probable that TdP can be attributed to quinidine in cases where it arises in patients (far more so than for other compounds tested), which is consistent with clinical experience and the known safety profile of quinidine.

A comparison of the results in this study with risk scores/categories from previous classifiers and databases (where available) is shown in electronic supplementary material, table S2. While it is hard to compare directly these scores, it can be seen that there is reasonable agreement across the different classification systems regarding which drugs are risky, such as quinidine and HAL, and which drugs pose less of a TdP risk, such as QUIN. This suggests that the qNet metric may capture to a reasonable degree relative differences between compounds in the cellular-level mechanisms that determine TdP. Nonetheless, some discrepancies are apparent. One reason for this is that our model predictions are highly sensitive to the free $C_{max}$ input. This is not ideal as effective free therapeutic plasma concentrations are not easily obtained from the literature. A final crucial point regarding the classifier is that risk category prediction does not necessarily rely on accurate prediction of APD prolongation [20] and indeed it has been tested for predictions without considering APD [28]; as such, here we use the same classifier to assess cardiac risk while making no claims about the accuracy of the degree of APD prolongation in the model. Further, while we have searched the literature for ion channel blocking effects of the listed compounds, we acknowledge that many compounds have active metabolites and that any potential impact of these upon APD has not been modelled in this analysis.

Patient risk factors can also predispose to cardiotoxicity, some of which may be more common in COVID-19, e.g. electrolyte imbalances, renal failure, drug interactions. COVID-19 also appears to affect the heart (e.g. myocarditis), which may additionally increase cardiotoxicity risk [46]. COVID-19 is also associated with acute kidney injury and electrolyte abnormalities [47]. Safety of drug–drug interactions (including combinations of anti-malarials) are an important consideration when being explored as a therapeutic in comorbid patients [48].

## 5. Conclusion

The safety profile of both CQ and HCQ are well-established. Since the SARS-CoV-2 outbreak, the ability of CQ/HCQ to inhibit certain coronaviruses has been explored. Although interest in HCQ use for acutely unwell COVID-19 patients has waned, there is a continued interest in potential use at symptom onset or as a prophylactic, with combination therapies with zinc and/or AZ under continued discussion in peer-reviewed and popular literature. In this study, we demonstrate an *in silico* safety assessment with uncertainty quantification based on the CiPA qNet torsade metric score. At a free $C_{max}$ of approximately 1.2 μM as seen in malaria treatment, HCQ would most likely be placed in the high-intermediate-risk category for TdP arrhythmia, whereas CQ was predicted to most likely lie in the intermediate-risk category at a free $C_{max}$ of approximately 0.7 μM. Combining HCQ with the antibacterial MOX, or the anti-malarial HAL was predicted to increase risk compared to administration of HCQ alone, increasing the degree of human ventricular APD prolongation at some or all concentrations investigated. Further clinical work will be required in order to assess the cardiac effects of HCQ at different doses as used in specific disease populations.

Data accessibility. The data underlying this article are available on Github in the following repository: https://github.com/CardiacModelling/risk-stratification-anti-malarials. All simulation data and codes required to run the simulations are freely available in Github: https://github.com/CardiacModelling/risk-stratification-anti-malarials and have been archived within the Zenodo repository: https://doi.org/10.5281/zenodo.4650300.

Authors' contributions. D.G.W.: designed experiments, conducted experiments, data analysis, contributed to manuscript writing. R.A.C: conducted experiments, contributed to manuscript writing. M.H.: contributed to experiments. X.H.S.C., N.J.W. and N.H.: contributed to manuscript writing and data interpretation. G.R.M. and R.-A.B.B.: conceived the idea, drafted the manuscript, contributed to data analysis and data interpretation.

Competing interests. We declare we have no competing interests.

Funding. This work was supported by the Wellcome Trust and Royal Society (grant no. 109371/Z/15/Z); the British Heart Foundation (grant nos. PG/18/4/33521 and FS/15/8/3115); the Wellcome Trust (grant no. 212203/Z/18/Z); and the BHF Centre of Research Excellence (grant no. RE/08/004). R.-A.B.B. and R.A.C. acknowledge support from a Sir Henry Dale Wellcome Trust and Royal Society fellowship, and R.-A.B.B. acknowledges support from The Returning Carers' Fund, Medical Sciences Division, University of Oxford and the British Heart Foundation. G.R.M. and D.G.W. acknowledge support from the Wellcome Trust via a Wellcome Trust Senior Research

Fellowship to G.R.M. N.H. acknowledges support from a British Heart Foundation intermediate fellowship, and from the BHF Centre of Research Excellence.

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
