## [Peer Review File · Royal Society Open Science]

Review History

RSOS-210235.R0 (Original submission)

Review form: Reviewer 1

Is the manuscript scientifically sound in its present form?

Yes

Are the interpretations and conclusions justified by the results?

Yes

Is the language acceptable?

Yes

Do you have any ethical concerns with this paper?

No

Have you any concerns about statistical analyses in this paper?

No

Recommendation?

Accept as is

Comments to the Author(s)

Authors replied to all my comments and clarified doubts. Manuscript is ready to be published. Good luck!

Decision letter (RSOS-210235.R0)

Dear Dr Burton:

I am pleased to inform you that your manuscript entitled "Cardiac TdP Risk Stratification Modelling of Anti-Infective Compounds including Chloroquine and Hydroxychloroquine" is now accepted for publication in Royal Society Open Science.

Please ensure that you send to the editorial office an editable version of your accepted manuscript, and individual files for each figure and table included in your manuscript. You can send these in a zip folder if more convenient. Failure to provide these files may delay the processing of your proof.

Additionally, at this stage, we ask that you please archive your GitHub code within the Zenodo repository: <https://guides.github.com/activities/citable-code/> <https://guides.github.com/activities/citable-code/> <https://guides.github.com/activities/citable-code/> <https://guides.github.com/activities/citable-code/>. By doing this, a formal, citable DOI will be associated with your data record, and an open license (CC-BY preferred) can be applied to your data. We can then update your data availability statement to read as:

All simulation data and codes required to run the simulations are freely available in the Github repository: <https://github.com/CardiacModelling/risk-stratification-anti-malarial>, and have been archived within the Zenodo repository: <https://doi.org/zenodo...> <https://doi.org/zenodo...> <https://doi.org/zenodo...> <https://doi.org/zenodo...> [ref number].

Please could you kindly reply to this email and provide the Zenodo DOI once you have done this? At the proofing stage of your manuscript, we would ask that you please then add the full citation of the Zenodo dataset to the end of your main References list at the end of your manuscript. Data references in Royal Society journals are in the Vancouver style, for example:

1. Torres-Campos I, Abram PK, Guerra-Grenier E, Boivin G, Brodeur J. 2016 Data from: A scenario for the evolution of selective egg colouration: the roles of enemy-free space, camouflage, thermoregulation, and pigment limitation. Dryad Digital Repository. (<https://doi.org/10.5061/dryad.5qt2k>).

COVID-19 rapid publication process:

We are taking steps to expedite the publication of research relevant to the pandemic. If you wish, you can opt to have your paper published as soon as it is ready, rather than waiting for it to be published the scheduled Wednesday.

This means your paper will not be included in the weekly media round-up which the Society sends to journalists ahead of publication. However, it will still appear in the COVID-19 Publishing Collection which journalists will be directed to each week (<https://royalsocietypublishing.org/topic/special-collections/novel-coronavirus-outbreak>).

If you wish to have your paper considered for immediate publication, or to discuss further, please notify openscience_proofs@royalsociety.org and press@royalsociety.org when you respond to this email.

Best regards,

on behalf of Dr Francois Fages (Associate Editor) and Professor Kevin Padian (Subject Editor).

Associate Editor Dr Francois Fages Comments to Author:

Dear Authors

It is my pleasure to inform you that your manuscript is accepted as is in RSOS.

Congratulations for this manuscript revision.

Best regards

Reviewer(s)' Comments to Author:

Reviewer: 1

Comments to the Author(s)

Authors replied to all my comments and clarified doubts. Manuscript is ready to be published. Good luck!
